# S³MAMBA: ARBITRARY-SCALE SUPER-RESOLUTION VIA SCALEABLE STATE SPACE MODEL

## ABSTRACT

Arbitrary scale super-resolution (ASSR) aims to super-resolve low-resolution images to high-resolution images at any scale using a single model, addressing the limitations of traditional super-resolution methods that are restricted to fixed-scale factors (e.g., $\times 2$, $\times 4$). The advent of Implicit Neural Representations (INR) has brought forth a plethora of novel methodologies for ASSR, which facilitate the reconstruction of original continuous signals by modeling a continuous representation space for coordinates and pixel values, thereby enabling arbitrary-scale super-resolution. Consequently, the primary objective of ASSR is to construct a continuous representation space derived from low-resolution inputs. However, existing methods, primarily based on CNNs and Transformers, face significant challenges such as high computational complexity and inadequate modeling of long-range dependencies, which hinder their effectiveness in real-world applications. To overcome these limitations, we propose a novel arbitrary-scale super-resolution method, called S³Mamba, to construct a scalable continuous representation space. Specifically, we propose a Scalable State Space Model (SSSM) to modulate the state transition matrix and the sampling matrix of step size during the discretization process, achieving scalable and continuous representation modeling with linear computational complexity. Additionally, we propose a novel scale-aware self-attention mechanism to further enhance the network's ability to perceive global important features at different scales, thereby building the S³Mamba to achieve superior arbitrary-scale super-resolution. Extensive experiments on both synthetic and real-world benchmarks demonstrate that our method achieves state-of-the-art performance and superior generalization capabilities at arbitrary super-resolution scales. The code will be publicly available.

## 1 INTRODUCTION

With the rapid advancement of digital imaging technology and computational photography, image super-resolution (SR) has become a significant research topic in computer vision and image processing (Gunturk et al., 2004; Zou & Yuen, 2011; Shi et al., 2013; Peng et al., 2024b; Conde et al., 2024; Ren et al., 2024). SR aims to reconstruct high-resolution (HR) images from low-resolution (LR) inputs to enhance visual quality. However, traditional factor-fixed SR methods (Lim et al., 2017; Zhang et al., 2018a; Liang et al., 2021; Chen et al., 2022b; Peng et al., 2024c;a) often can only upscale LR images by fixed magnification factors (Shi et al., 2016; Zhang et al., 2018b; Niu et al., 2020), such as ($\times 2$, $\times 3$, $\times 4$), which makes it difficult to meet the demands of real-world applications that require arbitrary magnification. Consequently, arbitrary-scale SR (ASSR) has been proposed and has garnered widespread attention, effectively achieving any SR scale using a single model.

In reality, our physical world is three-dimensional and continuous. To record the physical world, various imaging devices have been invented to discretize signals by capturing reflected photons from the real world to obtain observable digital images, as shown in Fig. 1 (a). The limited quality and resolution of sensors result in low-quality LR images. Therefore, the biggest challenge of ASSR is to learn the continuous signals of the real world from these discretized LR images (Chen et al., 2021; Lee & Jin, 2022; Cao et al., 2023). Numerous approaches have been proposed to achieve this. Among these, implicit neural representation (INR) stands out as the most prominent and effective.

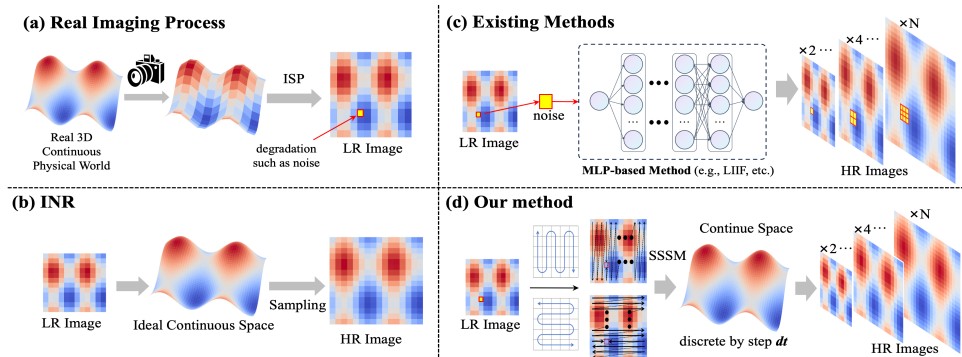

Figure 1: (a) During real-world imaging, the continuous 3D physical world is discretized into an image through cameras and ISPs, resulting in an LR image due to sensor resolution. (c) Existing MLP-based INR methods often use point-to-point learning, making them susceptible to degradation such as noise. Additionally, the limited receptive field of MLPs cannot construct a perfect continuous space, as shown in (b). In contrast, our method (d) leverages scalable SSM to better capture global historical information and, through scalable training, reconstructs a continuous space more effectively, achieving superior ASSR.

INR constructed a mapping from continuous pixel coordinates and the discredited low-resolution images to the continuous HR signal, achieving scalable SR, as shown in Fig. 1 (b).

Numerous INR-based ASSR methods have been proposed, achieving significant progress. For instance, LIIF (Chen et al., 2021) is the first to introduce INR into arbitrary-scale super-resolution, utilizing multi-layer perceptrons (MLP) to reconstruct continuous mappings for arbitrary-scale enlargement. This approach has achieved impressive visual results and garnered significant attention. Following this, LTE (Lee & Jin, 2022) and LINF (Yao et al., 2023) attempt to enhance performance by incorporating frequency domain information in the decoder. However, the limited receptive field and point-to-point learning approach of MLP make it difficult to capture contextual information, leading to challenges in constructing continuous HR images and making them susceptible to noise interference. This constrains the performance of INR in ASSR, as shown in Fig. 1 (c). Therefore, CiaoSR (Cao et al., 2023) and CLIT (Chen et al., 2023) utilized Transformers to model global information, significantly improving model performance. Although Transformers excel at modeling relationships among all tokens to capture contextual information, their self-attention mechanism incurs quadratic computational complexity. This quadratic increase in complexity with respect to input size makes them inefficient for real-world deployment. Therefore, there is an urgent need for an ASSR network capable of global modeling while maintaining high efficiency.

To address the aforementioned challenges, we propose an innovative ASSR method called $S^3$Mamba, which constructs a scalable continuous representation space, as shown in Fig. 1 (d). This approach introduces the State Space Model (SSM) into ASSR for the first time. We further propose a novel Scalable State Space Model (SSSM) to modulate the state transition matrix and sampling step size during discretization, thus achieving scalable and continuous representation modeling with linear computational complexity. Additionally, we develop an advanced scale-aware self-attention mechanism to enhance the network's ability to capture globally significant features across various scales. These innovations culminate in $S^3$Mamba, a versatile module that integrates seamlessly into various SR backbones, thus improving their efficacy at arbitrary scales. Comprehensive experiments on both real-world and popular synthetic benchmarks demonstrate our method's state-of-the-art(SOTA) performance, with superior generalization and continuous space representation capabilities in real-world scenarios. Our main contributions are as follows:

- We pioneer the introduction of SSM into arbitrary-scale super-resolution and propose the novel Scalable State Space Model. This model effectively modulates the state transition matrix and sampling step size during discretization, achieving scalable and continuous representation modeling with linear computational complexity.

- We develop the $S^3$Mamba, introducing an innovative scale-aware self-attention mechanism that incorporates the SSSM. This enhancement significantly boosts the network's ability to capture globally significant features across various scales, ensuring superior performance at arbitrary scales.

- Extensive experiments demonstrate that our method achieves the best performance on the popular DIV2K benchmark and exhibits the best performance and generalization capabilities on real-world COZ benchmarks.

## 2 RELATED WORK

### 2.1 ARBITRARY-SCALE SUPER-RESOLUTION

Different from traditional fixed-scale single image super-resolution (Dong et al., 2014; Ledig et al., 2017; Kim et al., 2016; Cavigelli et al., 2017; Zhang et al., 2021; Wang et al., 2018), Arbitrary-Scale Super-Resolution (ASSR) has the ability to enhance image quality and resolution across various scales, garnering significant attention in the fields of image processing and computer vision. For example, MetaSR first proposed a meta-upscale module to tackle this challenge (Hu et al., 2019). Inspired by the success of implicit neural representation (INR) in 3D shape reconstruction (Sitzmann et al., 2020; Chen & Zhang, 2019; Michalkiewicz et al., 2019; Gropp et al., 2020; Sitzmann et al., 2019; Mildenhall et al., 2021), the LIIF method employs MLPs to learn a continuous representation of the image. It takes continuous image coordinates and surrounding image features as input, outputting the RGB values at given coordinates. However, MLP has limitations in learning high-frequency components. LTE addresses this issue by effectively encoding image textures in the Fourier space. SRNO (Wei & Zhang, 2023) introduces neural operators to capture global relationships within the image. ITSRN (Yang et al., 2021) further innovatively proposes an implicit transformer based on INR structures to fully leverage screen image content. Cao *et al.* proposed CiaoSR as a continuous implicit attention network, that learns and integrates the weights of local features nearby, achieving the current SOTA performance. These methods provide diverse pathways and possibilities for achieving arbitrary-scale super-resolution. LMF (He & Jin, 2024) optimized image representation by reducing MLP dimensions and controls rendering intensity through modulation to reduce the computational cost of the upsampling module. COZ (Fu et al., 2024b) provided a benchmark for real-world scenarios, offering a dataset for arbitrary-scale super-resolution tasks captured in real scenes, along with a lightweight INR network. However, the aforementioned ASSR methods primarily utilize MLPs for point-to-point generation of high-resolution image pixels, which tends to overlook the intrinsic continuity within images. This oversight makes them susceptible to degradation artifacts, resulting in unrich detail and artifacts.

### 2.2 STATE SPACE MODELS

State Space Models (SSMs) were first developed in the 1960s for control systems (Kalman, 1960), providing a framework for modeling systems with continuous signal inputs. In recent times, the evolution of SSMs has facilitated their integration into the realm of computer vision (Zhu et al., 2024; Patro & Agneeswaran, 2024; Fu et al., 2024a; Chen et al., 2024). A prominent example is Visual Mamba, which introduced a residual VSS module and implemented four scanning directions. This innovation resulted in superior performance over ViT (Dosovitskiy et al., 2020), while maintaining a lower model complexity, thus garnering considerable attention (Guo et al., 2024; Qiao et al., 2024; Tang et al., 2024; Wang et al., 2024; Zhen et al., 2024; Li et al., 2024; Yan et al., 2024; Xiao et al., 2024). Notably, MambaIR (Guo et al., 2024) pioneers the use of SSMs in image restoration, boosting both efficiency and global perceptual capability. Despite these advances, the potential of the continuous representation modeling ability of SSM in arbitrary-scale super-resolution tasks remains underexplored. Therefore, we propose a novel scalable State Space State, which leverages the continuous state space of SSMs to enhance the network's capability in continuous representation, thereby achieving high-quality continuous arbitrary-scale super-resolution.

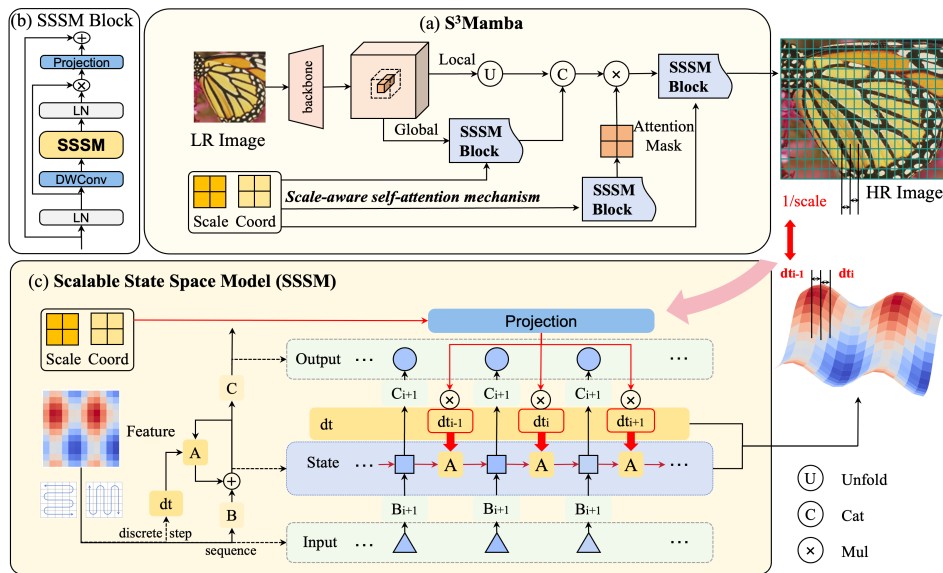

Figure 2: (a) Illustration of the proposed $S^3$Mamba framework. (b) The SSSM Block consists of the SSSM, along with multiple instance normalization layers, depthwise convolution (DWConv), and projection layers. (c) The Scalable State Space Model (SSSM) is proposed to modulate the state transition matrix and the sampling matrix of step size during the discretization process, achieving scalable and continuous representation modeling with linear computational complexity.

## 3 PRELIMINARY AND MOTIVATION

The three-dimensional, continuous physical world is recorded by cameras that convert reflected photons into digital images (Son et al., 2012), as shown in Fig. 1 (a). However, limitations in CMOS and CCD sensor technology result in LR images, failing to meet consumers' demands for higher resolution and better quality (Wang et al., 2020; Chen et al., 2022a). Image super-resolution techniques have been developed to generate HR images from LR counterparts. Unlike traditional fixed-scale methods, ASSR aims to reconstruct the original continuous scene, generating HR images at any resolution. The main challenge of ASSR is learning continuous signals from discretized data (Liu et al., 2024; Chen et al., 2021), as shown in Fig. 1 (b). The Implicit Neural Representation stands out as the most prominent and effective in ASSR. By learning the mapping from pixel coordinates to pixel values, INR is able to generate scalable, high-quality HR images, as formulated:

$$F_{LR} = \Psi(LR) \tag{1}$$

$$HR_{(i,j)}^{RGB} = \phi(F_{LR}, coord_{(i,j)}, scale) \tag{2}$$

where, $F_{LR}$ represents the features of the low-resolution image $LR$, and $\Psi$ denotes the feature extractor. $coord$ represents the coordinates location, $scale$ indicates the magnification factor, and $HR^{RGB}$ denotes the high-resolution RGB image. The goal of INR is to learn a continuous function $\phi$ that maps coordinates and images to continuous signals, effectively mapping different scales of the same scene into a unified continuous representation space. Various INR methods, like LIIF and LTE, use a MLP for ASSR, but MLPs' limited receptive field and point-to-point approach ignore contextual and historical data, leading to vulnerability to noise and poor continuous representation capability, as shown in Fig. 1 (c). An intuitive approach is to introduce Transformers to capture global information, as seen in methods like CiaoSR. While Transformers effectively capture context, their quadratic computational complexity makes them impractical for real-world applications. More detailed analyses and comparisons are provided in the Appendix.

## 4 METHOD

To reconstruct a scalable continuous representation space, we propose a novel ASSR method called $S^3$Mamba, as shown in Fig. 2 (a). This approach leverages the Scalable State Space Model (SSSM) to

adaptively capture global and scale-dependent features, ensuring consistent continuous representations across varying scales. Additionally, our innovative scale-aware self-attention mechanism is introduced to enhance the network's ability to perceive globally significant features at different scales, thereby reconstructing high-quality HR images efficiently and effectively.

## 4.1 PROPOSED SCALABLE STATE SPACE MODEL

To capture global historical information without incurring significant computational overhead, we turn our attention to state space models. Benefiting from the linear complexity and global modeling capacity of SSM, we introduce the SSM into ASSR for the first time. Let's briefly review SSM. The latest advances in structured state space models (S4) are largely inspired by continuous linear time-invariant systems, which map input $x(t)$ to output $y(t)$ through an implicit latent state $h(t) \in \mathbb{R}^N$ (Guo et al., 2024). This system can be represented as a linear ordinary differential equation:

$$\dot{h}(t) = Ah(t) + Bx(t), \quad y(t) = Ch(t) + Dx(t). \tag{3}$$

where $N$ is the state size, $A \in \mathbb{R}^{N \times N}$, $B \in \mathbb{R}^{N \times 1}$, $C \in \mathbb{R}^{1 \times N}$, and $D \in \mathbb{R}$. To adapt to digital information processing, the continuous function of the state space model in Eq. 3 is discretized into a sequence analysis model. Specifically, the state space model uses a zero-order hold as follows:

$$\overline{A} = \exp(\Delta A), \quad \overline{B} = (\Delta A)^{-1}(\exp(\Delta A) - I)\Delta B, \tag{4}$$

In this process, the sampling interval $\Delta$ determines the arrangement of discrete signals, so within the SSM, $\Delta$ dictates the correlation and association between adjacent inputs. Finally, we arrive at the discrete state space representation, as shown in the following equations:

$$h_k = \overline{A}h_{k-1} + \overline{B}x_k, \quad y_k = Ch_k + Dx_k, \tag{5}$$

In the traditional state space model, $\Delta$ is determined solely by the current input, making it well-suited for scale-invariant vision tasks. However, because the actual physical distance between adjacent pixels varies with different scales of the same scene, the correlation and association between adjacent pixels also change with scale. An INR model trained solely on a traditional SSM may fail to capture these scale-dependent patterns, resulting in different continuous representations for different scales of the same scene. This is inconsistent with the fundamental goal of INR.

To address this issue, we propose a novel Scalable SSM, which incorporates scale and continuous coordinate information into the state space equations of the state space model and adjusts $\Delta_{x_k}$ to achieve scale awareness. Specifically, we use a learnable MLP layer to input the scale, generating a scale modulation factor for each time step, which is introduced into the current $\Delta_{x_k}$, as formulated:

$$\Delta_{x_k} = \omega(x_k), \Delta_{x_k}^{scale} = \sigma(scale, coord_{x_k}),$$
$$\Delta'_{x_k} = \Delta_{x_k} \cdot \Delta_{x_k}^{scale}. \tag{6}$$

where $\omega$ and $\sigma$ represent multilayer perceptron layers. This approach allows the SSSM to adaptively adjust the interaction patterns of adjacent points at different scales. This ensures consistency in network outputs for the same input across various scales, allowing our ASSR model to maintain a consistent continuous representation space when handling data of different output sizes.

Furthermore, in the original state space equations, the parameter matrix $B$ represents the mapping pattern from the input to the state space. It is directly determined by the current input to produce $B_{x_k}$, which can still prevent the SSM-based upsampling module from effectively capturing continuous space representation methods at different scales. Therefore, we follow Eq. 6 to transform the same process to matrix $B_{x_k}$ into $B'_{x_k}$, allowing it to better perceive the mapping equations across different scales. This ensures that the state space model can adapt to any magnification level. The above process can be formulated as:

$$B_{x_k}, C_{x_k}, \Delta_{x_k} = \omega(x_k),$$
$$B_{x_k}^{scale}, \Delta_{x_k}^{scale} = \sigma(scale, coord_{x_k}), \tag{7}$$
$$\Delta'_{x_k} = \Delta_{x_k} \cdot \Delta_{x_k}^{scale}, B'_{x_k} = B_{x_k} \cdot B_{x_k}^{scale}.$$

Then, the discretization process of our Scalable State Space Model (SSSM) can be formulated as :

$$\overline{A}'_{x_k} = \exp(\Delta'_{x_k}A),$$
$$\overline{B}'_{x_k} = (\Delta'_{x_k}A)^{-1}(\exp(\Delta'_{x_k}A) - I)\Delta'_{x_k}B'_{x_k}, \tag{8}$$

Finally, the discretized state space equations of our SSSM can be represented by Eq. 5. Through the aforementioned design, we follow to (Zhu et al., 2024) to construct the SSSM block to adeptly capture scale variations, as shown in Fig. 2 (b) and (c). This allows LR images, sampled at different scales within a unified continuous scene, to be represented within a single continuous space. This capability facilitates the construction of an enhanced continuous space, yielding HR images across arbitrary scales that are visually pleasing and rich in detail.

## 4.2 Proposed S³Mamba

Further, to integrate global information and strengthen the scale-invariant perception capability of the feature space, we employ the SSSM as an efficient global feature extraction method to supplement global information. We also propose a novel scale-aware self-attention mechanism to further enhance the network's ability to perceive globally important features at different scales, as illustrated in Fig. 2. Specifically, for a LR image, we first extract its features through a backbone, obtaining $F_{LR}$. Additionally, we follow (Cao et al., 2023) by using the Unfold operation to aggregate local features and obtain local information $F_{LR}^{local}$. The SSSM is utilized to extract global features $F_{LR}^{global}$. These are combined to form a new fused feature for subsequent representation learning, as shown in the following equations:

$$F_{LR}^{local}, F_{LR}^{global} = U(F_{LR}), SSSM(F_{LR}),$$
$$F_{fusion} = concat(F_{LR}^{local}, F_{LR}^{global}). \tag{9}$$

where $U$ represents the unfold operation to capture local features. In addition, considering the inconsistency in feature distribution across different scales, we propose a scale-aware self-attention mechanism to enhance the network's focus on the feature representation at the current scale. This mechanism aims to learn a feature-independent global mapping pattern under various transformation modes. Specifically, we input $coord_{HR}$ and $scale$ into the SSSM to generate a global self-attention map $\alpha_{weight}$. This attention map, guided by the current scale and coordinates, adaptively refines HR feature $F_{HR}$, ultimately yielding $RGB_{HR}$. The process is illustrated by the following equations:

$$\alpha_{weight} = SSSM(coord_{HR}, scale),$$
$$F_{HR}' = SSSM(\alpha_{weight} \cdot F_{HR}), \tag{10}$$
$$RGB_{HR} = SSSM(F_{HR}').$$

Finally, we build a simple yet efficient ASSR architecture called S³Mamba, as illustrated in Fig. 2 (a).

## 5 Experiment and Analysis

### 5.1 Experiments Setting

**Datasets.** To evaluate on the real-world ASSR task, we follow the training and test set from COZ, which consists of 153 training scenes comprising a total of 9,019 images and 37 testing scenes at 2K resolution. For the conventional ASSR task based on bicubic degradation, following previous work(LIIF, LTE, Ciao *et al.*), we also use the commonly employed synthetic DIV2K dataset as the training set, which consists of 800 HR images in 2K resolution for training by the bicubic degradation model. For testing, we evaluate the performance in the DIV2K validation set with 100 HR images.
**Evaluation metrics.** Following previous work, we use PSNR and SSIM (Wang et al., 2004) to evaluate the quality of the HR images generated. All quantitative metrics in our experiments were consistently evaluated in the RGB color space for both benchmark datasets, DIV2K (Agustsson & Timofte, 2017) and COZ.

**Implementation details.** Following previous works, we adopt the same way to generate paired images for training on the synthetic DIV2K data set. Initially, we crop image patches of size $96s \times 96s$ as ground truth (GT), where $s$ is a scaling factor sampled from a uniform distribution U(1, 4). Then, we use bicubic downsampling to generate the corresponding LR images. We employ existing SR models, such as EDSR and RDN, as the backbones to evaluate various arbitrary-scale upsampling methods. Adam is used as the optimizer, with the initial learning rate set to 1e-4 and decaying by a factor of 0.5 every 200 epochs. During training, our method follows previous works, training for

Table 1: Quantitative comparison of ASSR methods on the real-world COZ validation set (PSNR in dB / SSIM). Bold indicates best performance; underlined indicates second-best. "Out-of-scale" denotes evaluation on scales absent from model training data.

| Backbones | Methods | In-scale | | | Out-of-scale | | |
|---|---|---|---|---|---|---|---|
| | | ×3 | ×3.5 | ×4 | ×5 | ×5.5 | ×6 |
| EDSR | MetaSR | 26.55/0.767 | 25.62/0.752 | 25.17/0.740 | 24.31/0.720 | 23.93/0.711 | 23.25/0.678 |
| | LIIF | 26.61/0.767 | 25.76/0.752 | 25.16/0.741 | 24.32/0.721 | 24.01/0.711 | 23.23/0.679 |
| | LTE | 26.55/0.767 | 25.71/0.752 | 25.15/0.740 | 24.37/0.720 | 24.05/0.712 | 23.26/0.679 |
| | LINF | 26.53/0.762 | 25.66/0.750 | 25.10/0.737 | 24.29/0.719 | 23.99/0.711 | 23.21/0.677 |
| | SRNO | 26.59/0.766 | 25.70/0.752 | 25.15/0.741 | 24.31/0.722 | 24.05/0.712 | 23.25/0.680 |
| | LIT | 26.58/0.766 | 25.71/0.753 | 25.16/0.741 | 24.35/0.721 | 24.00/0.712 | 23.19/0.679 |
| | CiaoSR | 26.56/0.770 | 25.65/0.755 | 25.13/0.746 | 24.31/0.725 | 23.96/0.721 | 23.23/0.709 |
| | LMI | 26.66/0.768 | 25.78/0.752 | 25.22/0.741 | 24.39/0.722 | 24.08/0.713 | 23.29/0.680 |
| | **Ours** | **26.71/0.773** | **25.84/0.755** | **25.27/0.746** | **24.39/0.726** | **24.09/0.723** | **23.34/0.709** |
| RDN | MetaSR | 26.65/0.767 | 25.80/0.752 | 25.22/0.740 | 24.39/0.720 | 24.09/0.711 | 23.31/0.678 |
| | LIIF | 26.69/0.766 | 25.83/0.752 | 25.23/0.740 | 24.39/0.718 | 24.13/0.711 | 23.28/0.679 |
| | LTE | 26.64/0.767 | 25.74/0.752 | 25.17/0.740 | 24.40/0.719 | 24.10/0.709 | 23.28/0.676 |
| | LINF | 26.60/0.762 | 25.73/0.750 | 25.15/0.737 | 24.32/0.719 | 24.03/0.711 | 23.28/0.677 |
| | SRNO | 26.67/0.766 | 25.73/0.752 | 25.19/0.741 | 24.40/0.722 | 24.09/0.712 | 23.28/0.680 |
| | LIT | 26.66/0.766 | 25.79/0.753 | 25.19/0.741 | 24.36/0.721 | 24.03/0.712 | 23.25/0.679 |
| | CiaoSR | 26.61/0.772 | 25.76/0.756 | 25.22/0.746 | 24.38/0.727 | 24.06/0.721 | 23.36/0.710 |
| | LMI | 26.74/0.769 | 25.86/0.753 | 25.30/0.742 | 24.48/0.723 | 24.14/0.714 | 23.37/0.682 |
| | **Ours** | **26.74/0.777** | **25.92/0.760** | **25.34/0.749** | **24.50/0.728** | **24.15/0.724** | **23.39/0.710** |

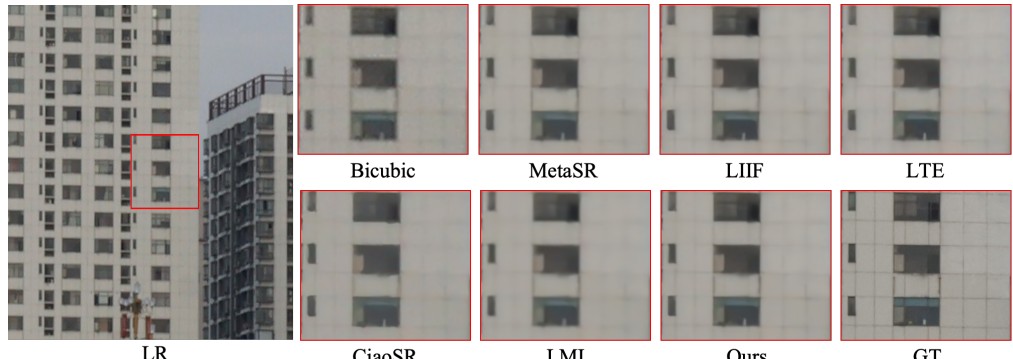

Figure 3: Visual comparison with existing methods on COZ dataset × 3. Zoom in for a better view.

a total of 1000 epochs under L1 loss. The total batch size is set to 32, utilizing a total of 8 V100 GPUs. For real-world ASSR datasets, we follow the training and testing setting of COZ to ensure fair evaluation, while the total batch size is set to 128. We retrain some unreported models in COZ with same training configurations to ensure comparability.

**Compared methods.** To demonstrate the superiority of our model, we conduct a performance comparison against eight SOTA or popular models: MetaSR, LIIF, LTE, LINF, SRNO, (C)LIT, CiaoSR, and LMI(COZ) under two popular backbones EDSR (Lim et al., 2017) and RDN (Zhang et al., 2018b).

## 5.2 QUANTITATIVE AND QUALITATIVE RESULTS

In Tab. 1 and Tab. 2, we conduct a comparative experiment on COZ and DIV2K to assess its performance against existing methods, to demonstrate the superiority of our method.

**Results on the real-world dataset.** As demonstrated in Tab. 1, our method achieves significant superiority on the COZ dataset, with marked improvements in both PSNR and SSIM metrics. Crucially, our approach substantially outperforms LMI in SSIM and CiaoSR in PSNR, underscoring its consistent advantage.This demonstrates that our method is capable of better reconstructing continuous image representations, thereby enhancing image detail performance. Additionally, we conduct a

Table 2: Quantitative comparison of ASSR methods on the synthetic DIV2K validation set (PSNR in dB). Bold indicates best performance; underlined indicates second-best. "Out-of-scale" denotes evaluation on scales absent from model training data.

| Backbones | Methods | Params/[K] | FLOPs/[G] | In-scale | | | Out-of-scale | | |
|---|---|---|---|---|---|---|---|---|---|
| | | | | ×2 | ×3 | ×4 | ×6 | ×12 | ×18 |
| | Bicubic | - | - | 31.01 | 28.22 | 26.66 | 24.82 | 22.27 | 21.00 |
| EDSR | baseline | - | - | 34.55 | 30.90 | 28.94 | - | - | - |
| | LMI | 85.8 | 15.22 | 34.59 | 30.90 | 28.94 | 26.69 | 23.68 | 22.18 |
| | MetaSR | 445.1 | 15.21 | 34.64 | 30.93 | 28.92 | 26.61 | 23.55 | 22.03 |
| | LIIF | 338.8 | 47.50 | 34.67 | 30.96 | 29.00 | 26.75 | 23.71 | 22.17 |
| | LTE | 482.3 | 27.77 | 34.72 | 31.02 | 29.04 | 26.81 | 23.78 | 22.23 |
| | ITSRN | 630.2 | 88.13 | 34.71 | 30.95 | 29.03 | 26.77 | 23.71 | 22.17 |
| | CLIT | 5203.8 | 122.88 | 34.81 | 31.12 | 29.15 | 26.92 | 23.83 | 22.29 |
| | SRNO | 774.3 | 27.11 | 34.85 | 31.11 | 29.16 | 26.90 | 23.84 | 22.29 |
| | CiaoSR | 1395.1 | 152.28 | 34.91 | **31.15** | 29.23 | 26.95 | 23.88 | **22.32** |
| | **Ours** | 771.3 | 86.94 | **34.93** | 31.13 | **29.24** | **26.97** | **23.89** | **22.32** |
| RDN | baseline | - | - | 34.94 | 31.22 | 29.19 | - | - | - |
| | LMI | 85.8 | 15.22 | 34.74 | 31.03 | 29.07 | 26.81 | 23.79 | 22.29 |
| | MetaSR | 445.1 | 15.21 | 35.00 | 31.27 | 29.25 | 26.88 | 23.73 | 22.18 |
| | LIIF | 338.8 | 47.50 | 34.99 | 31.26 | 29.27 | 26.99 | 23.89 | 22.34 |
| | LTE | 482.3 | 27.77 | 35.04 | 31.32 | 29.33 | 27.04 | 23.95 | 22.40 |
| | ITSRN | 630.2 | 88.13 | 35.09 | 31.36 | 29.38 | 27.06 | 23.93 | 22.36 |
| | CLIT | 5203.8 | 122.88 | 35.10 | 31.39 | 29.39 | 27.12 | 24.01 | 22.45 |
| | SRNO | 774.3 | 27.11 | 35.16 | 31.42 | 29.42 | 27.12 | 24.03 | 22.46 |
| | CiaoSR | 1395.1 | 152.28 | 35.15 | **31.42** | 29.45 | 27.16 | 24.06 | 22.48 |
| | **Ours** | 771.3 | 86.94 | **35.17** | 31.40 | **29.47** | **27.17** | **24.07** | **22.50** |

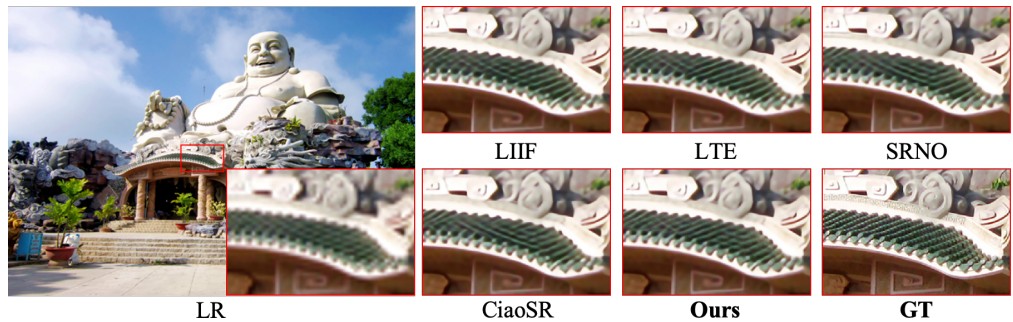

Figure 4: Visual comparison with existing methods on DIV2K dataset × 4. Zoom in for a better view.

visual comparison of the COZ dataset in Fig. 3. It is evident that compared with existing methods, our method more effectively removes degradation artifacts in real-world scenarios, reconstructing the details and textures of SR images closer to GT. This demonstrates that our method demonstrates stronger robustness against complex real-world degradations compared to conventional approaches. More comparison is available in the Supplement.

**Results on the DIV2K dataset.** As evidenced in Tab. 2, our method demonstrates superior performance across most scenarios, outperforming existing approaches at standard scales (×2, ×4, *etc.*) and dominating out-of-scale evaluations. While CiaoSR achieves comparable accuracy, our solution reduces computational complexity by nearly 50%. Visual comparisons on DIV2K shown in Fig. 4 further corroborate these advantages.It can be observed that the texture details reconstructed by our method are closer to the ground truth (GT), whereas other methods, such as the existing SOTA method CiaoSR, tend to produce artifacts. This demonstrates the superior visual performance of our approach. More visual comparison is available in the Supplement.

| PSNR | SSSM | SSM | linear attn | Convs | MLP |
|------|------|------|------|------|------|
| ×2 | **34.93** | 34.85 | 34.84 | 34.77 | 34.78 |
| ×4 | **29.24** | 29.17 | 29.15 | 29.06 | 29.09 |

Table 3: Performance comparison of different based models.

## 5.3 ABLATION STUDY

In this section, we conduct ablation studies to evaluate the effectiveness of core ideas of our method. We focus on two components: (a) Scalable State Space Model (SSSM) and (b) key elements in S³mamba, global feature extraction (GFE) and scale-aware self-attention (SFAtt). We use the EDSR baseline to validate their effectiveness on the DIV2K dataset.

**Effectiveness of SSSM.** To validate the effectiveness of the proposed SSSM module, we perform ablation experiments by replacing it with alternative components of equivalent linear complexity. Compared to global-receptive modeling approaches (SSM and Linear Attention) and local-receptive operations (Conv and MLP), our SSSM demonstrates superior performance, in Tab. 3. The significant performance gap between local-receptive and global-receptive variants highlights the necessity of integrating long-range dependencies during upsampling. Notably, SSSM consistently outperforms both SSM and Linear Attention among global-receptive approaches. This advantage comes from the unique explicit scale-conditioned state modulation of SSSM combined with spatially aware context aggregation, which jointly addresses the critical requirements of ASSR, preserving global consistency while recovering high-frequency textures across diverse magnification factors.

| GFE | SFAtt | ×2 | ×3 | ×4 |
|------|------|------|------|------|
| ✗ | ✗ | 34.71 | 30.98 | 29.06 |
| ✗ | ✓ | 34.78 | 31.03 | 29.12 |
| ✓ | ✗ | 34.85 | 31.09 | 29.19 |
| ✓ | ✓ | **34.93** | **31.13** | **29.24** |

| model | GFE | SFAtt |
|------|------|------|
| **SSSM** | **29.24** | **29.24** |
| Convs | 29.12 | 29.20 |
| MLP | 29.13 | 29.20 |
| None | 29.12 | 29.19 |

(a) PSNR comparison of GFE and SFAtt on DIV2K.      (b) PSNR with different core models on DIV2K ×4.

Table 4: Ablation of GFE and SFAtt

**Effectiveness of GFE and SFAtt.** Our experiments systematically validate the design of S³mamba dual branches as shown in Tab. 4. The module removal study confirms the necessity of both GFE and SFAtt: excluding GFE degrades feature representation due to insufficient global context, while removing SFAtt reduces scale adaptability by disabling position-aware analysis. Further component replacement tests (substituting global operations with local Conv / MLP counterparts) reveal that GFE's effectiveness stems specifically from its global information integration, which enriches local upsampling by resolving ambiguities across distant regions. Similarly, localized SFAtt implementations fail to decode scale-aware positional patterns due to constrained receptive fields.This dual validation reveals: GFE uniquely enhances local upsampling through global feature synthesis, while SFAtt enables scale-robust reconstruction via holistic position encoding analysis.

Additional experimental results can be found in the Appendix.

## 6 CONCLUSION

In this paper, we propose a novel SSSM that modulates the state transition and sampling matrices during the discretization process, achieving scalable and continuous representation modeling with linear computational complexity. Additionally, we develop a novel scale-aware self-attention mechanism to further enhance the network's ability to perceive globally significant features across various scales. The S³Mamba is designed for constructing scalable continuous representation spaces, enabling the reconstruction of arbitrary-scale high-resolution images with rich detail. Extensive experiments on both synthetic and real-world benchmarks demonstrate that our method not only achieves state-of-the-art results but also exhibits remarkable generalization capabilities, paving the new way for arbitrary-scale super-resolution.

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

---

**Algorithm 1** SSSM

---

1: **Input:** $x$ : (B, C, H, W) **Output:** $y$ : (B, C, H, W)
2: $x$ : (B, H*W, 4, C) $\leftarrow$ scan($x$)
3: $\boldsymbol{A}, \boldsymbol{D} \leftarrow$ Params
4: $\boldsymbol{C} \leftarrow s_C(x)$
5: $\boldsymbol{B} \leftarrow s_B(x) \cdot s_{Bscale}(scale, coord)$
6: $\boldsymbol{\Delta} \leftarrow \tau_{\boldsymbol{\Delta}}(\text{Params} + s_{\boldsymbol{\Delta}}(x) \cdot s_{\boldsymbol{\Delta}\text{scale}}(scale, coord))$
7: $\bar{\boldsymbol{A}}, \bar{\boldsymbol{B}} \leftarrow$ discretize($\boldsymbol{\Delta}, \boldsymbol{A}, \boldsymbol{B}$)
8: $y$ : (B, H*W, 4, C) $\leftarrow$ SSM($\bar{\boldsymbol{A}}, \bar{\boldsymbol{B}}, \boldsymbol{C}, \boldsymbol{D}$)($x$)
9: $y$ : (B, 4, C, H, W) $\leftarrow$ reshape($y$)
10: $y$ : (B, C, H, W) $\leftarrow$ mean($y$)
11: **return** $y$

---

# A APPENDIX

(a) Comparison of different computational architectures

|  | MLP | Transformer | RNN | SSM |
|---|---|---|---|---|
| Receptive Field | Local | Global | Near-global | **Near-global** |
| Computation Complexity | $O(n)$ | $O(n^2)$ | $O(n)$ | $\mathbf{O}(n)$ |
| Parallel | True | True | False | **True** |

(b) LPIPS tested on scale $\times 4$

| COZ | **Ours** | CiaoSR | LMI | LIIF |
|---|---|---|---|---|
| LPIPS↓ | **0.5211** | 0.5271 | 0.5899 | 0.5394 |

| DIV2K | **Ours** | CiaoSR | LTE | LIIF |
|---|---|---|---|---|
| LPIPS↓ | **0.2590** | 0.2593 | 0.2655 | 0.0.2670 |

Table 5: Ablation of GFE and SFAtt

**Organization.** In this part, we organize the Appendix as follows. In Section 1, we provide a comparison of the calculated effects of the RNN, Transformer, and SSM method. In Section 2, we provide the implementation details of the SSSM Block and the pseudocode of the proposed SSSM structure. In Section 3, we compare the computational efficiency of our method with other methods and provide additional results to demonstrate the effectiveness of our approach. In Section 4, we provide more visual effect contrasts as well as the user study results.

## A.1 ANALYSIS OF THE STATE SPACE MODEL

According to prior studies (Dao & Gu, 2024; Gu & Dao, 2023; Guo et al., 2024; Liu et al., 2021; Liang et al., 2021; Zhu et al., 2024), the State Space Model (SSM) architecture bridges key characteristics of both Transformer and Recurrent Neural Network (RNN) models while addressing their inherent limitations. Unlike the Transformer, which employs a global attention mechanism allowing every input point to interact with all others, SSM organizes inputs into a directed sequence and restricts the search scope to historical inputs stored selectively in the state space. This design reduces computational complexity and simultaneously maintains the ability to capture sequential dependencies. Furthermore, the dynamic evolution of the state space introduces a forgetting mechanism, ensuring efficient use of memory and computational resources.

Compared to RNN, SSM similarly processes historical information sequentially. However, RNN suffer from limited parallelization capability due to their inherently sequential computation, making them less practical for large-scale tasks. SSM overcomes this limitation by leveraging a State Space Model (SSM), which enables simultaneous computation of all outputs. This design allows SSM to maintain the sequential modeling strengths of RNN while achieving high parallel efficiency comparable to the Transformer.

SSM's key advantages lie in its balance between computational efficiency and modeling capacity, as shown in Tab. 5a. Narrowing the search scope to past inputs significantly reduces computational costs compared to the Transformer. Meanwhile, the SSM architecture ensures scalability and parallelism, effectively overcoming the limitations of RNN. This positions SSM as a powerful and efficient alternative for tasks that require both sequential dependency modeling and computational scalability.

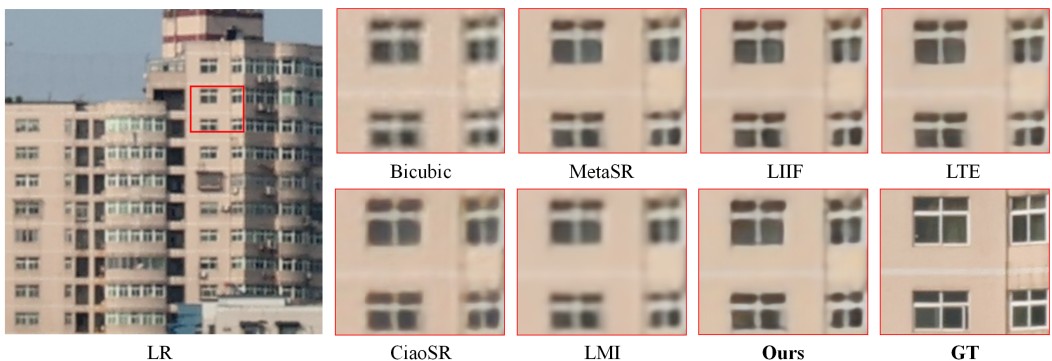

Figure 5: Visual comparison with existing methods on the COZ dataset $\times$ 3.5.

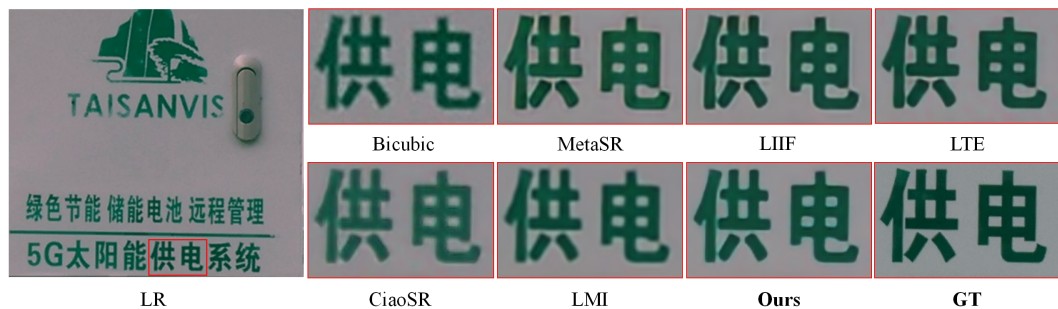

Figure 6: Visual comparison with existing methods on the COZ dataset $\times$ 4.

## A.2 Details of the proposed SSSM Block

The core architecture of the proposed SSSM block is grounded in the principles of the State Space Model (SSM) (Wang et al., 2023) to construct the SSSM framework. As elaborated in the main text, the SSM structure is characterized by five coefficient matrices, namely $A$, $B$, $C$, $D$, and $\Delta$. Following the methodology of SSM(S6) (Gu & Dao, 2023), we directly parameterize $A$ and $D$ to explicitly facilitate the implementation of forgetting mechanisms and skip connections, respectively. Similarly, the matrices $B$, $C$, and $\Delta$ are adaptively generated based on the input features, enabling effective dynamic state filtering. A key distinction lies in leveraging scale information to further inform and refine this generation process, with details shown in Algorithm 1. As discussed in the preceding section, we integrate results derived from multiple scanning strategies to enhance the structure's global contextual awareness and perception capabilities.

## A.3 More Results

### A.3.1 LPIPS

To provide comprehensive empirical validation of our method's perceptual superiority, we conducted extensive comparative analyses using the Learned Perceptual Image Patch Similarity (LPIPS) metric across two benchmark datasets: COZ and DIV2K. As quantitatively demonstrated in Table 5b, our approach achieves consistently lower LPIPS scores (indicating better perceptual alignment with human vision) compared with seven state-of-the-art methods, including CiaoSR, LTE, and LIIF. This result substantiates that our method's architectural innovations fundamentally enhance texture preservation and structural coherence under diverse degradation conditions.

### A.3.2 Visual Results

The visual results on the COZ dataset with real-world degradations are presented in Figures 5 and 6. As shown in Figure 5, in challenging scenarios characterized by significant noise and complex degradations, our method produces outputs with sharper edges and smoother regions. This highlights

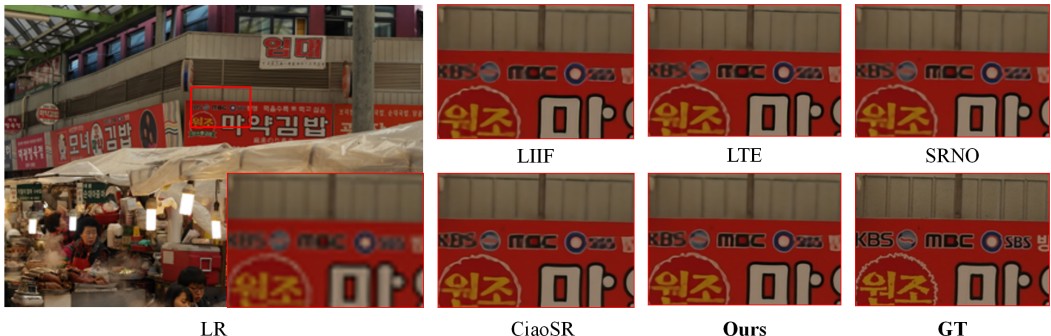

Figure 7: Visual comparison with existing methods on the DIV2K dataset $\times$ 4.

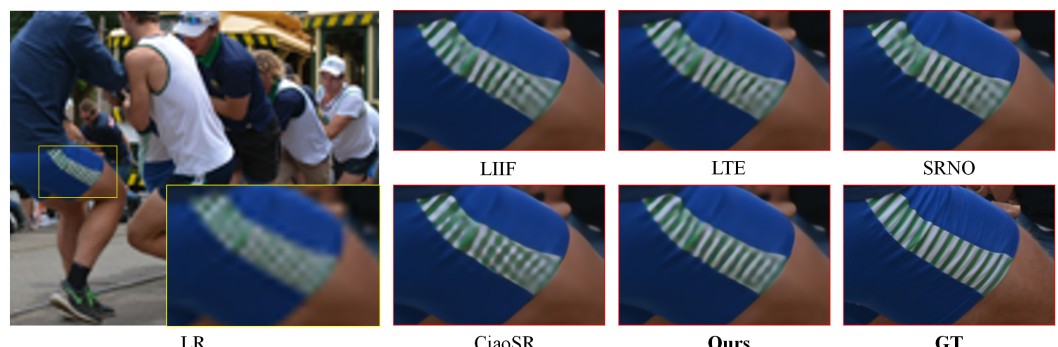

Figure 8: Visual comparison with existing methods on the DIV2K dataset $\times$ 8.

the efficacy of the proposed SSSM structure in effectively mitigating severe noise and addressing the complexities of real-world degradation. Additionally, Figure 6 demonstrates that, for regular textures with simpler structures, our method introduces fewer artifacts and achieves more uniform and consistent smooth regions. This indicates that the SSSM structure effectively leverages contextual information across the image, ensuring coherence and consistency in global pixel distribution.

Furthermore, the visual results on the DIV2K dataset under bicubic degradation are depicted in Figures 7 and 8. For textures with blurred structural details, our method delivers results that align more closely with the ground truth (GT) structure, demonstrating superior detail preservation and structural fidelity. This reflects the superior structural stability and reconstruction fidelity of our approach. These observations collectively underline the robustness of the SSSM structure in

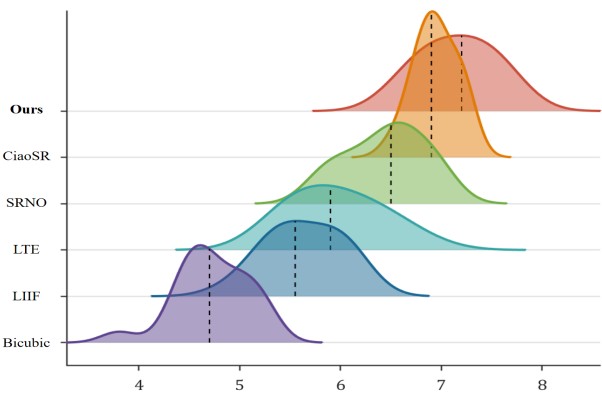

Figure 9: User study on the DIV2K dataset.

addressing diverse degradation scenarios, ensuring both local detail refinement and global structural consistency.

## A.4 USER STUDY

We selected 50 images from the test set of the DIV2K and COZ dataset and invited 20 participants to score each scene individually. The average score for each model was calculated, and the final statistical results are shown in figure 9. Clearly, the results generated by our method are visually superior, thanks to the global consistency constraint and structural stability of our model.

## A.5 USE OF LARGE LANGUAGE MODELS

Large Language Models (LLMs) were used only for grammar checking and text polishing. All research ideas, methods, and analyses are solely by the authors.

