# OpenReview forum: "$\text{S}^{3}$Mamba: Arbitrary-Scale Super-Resolution via Scaleable State Space Model"
_ICLR.cc/2026/Conference — ICLR 2026 Conference Withdrawn Submission_

### Official Review · Reviewer_QPgJ · 2025-10-28

**Soundness:** 2
**Presentation:** 2
**Contribution:** 2
**Rating:** 4
**Confidence:** 4

**Summary:**

The paper proposes a novel method called S3Mamba for arbitrary-scale super-resolution (ASSR). The method introduces a Scalable State Space Model (SSSM) that modulates state transition and sampling matrices during discretization, achieving scalable and continuous representation modeling with linear computational complexity. It also incorporates a scale-aware self-attention mechanism to improve the model's ability to capture global features at varying scales. Extensive experiments demonstrate that S3Mamba outperforms existing ASSR methods in synthetic and real-world settings, showing superior performance and generalization.

**Strengths:**

1. The SSSM considers the varying physical distances between adjacent pixels at different scaling factors, making it highly suitable for the ASSR task.
2. The scale-aware self-attention mechanism provides an innovative solution to enhance the model's ability to perceive global features across different scales.
3. The experimental results show that S3Mamba outperforms the comparison methods in most cases.

**Weaknesses:**

1. The ablation study lacks a comparison between the traditional SSM and the proposed scalable SSM, so we cannot determine whether the "Scalable" design leads to performance improvements. If the regular SSM can achieve good results, the novelty of this module would be reduced accordingly.
2. The authors only provide results where the model is trained and tested on the same dataset, lacking experiments demonstrating the model's generalization ability (e.g., training on DIV2K and testing on Urban100).
3. The visual comparison does not include "Out-of-scale" results (e.g., x6).
4. The title of Table 1 mentions that an underline represents second-best, but I don't see any underlines in the table.

**Questions:**

1. Most comparison methods report results at very large scales in their papers (e.g., x24, x30). Does S3Mamba still perform well at these scales? Also, why did the authors only provide results up to x6 on the COZ dataset?
2. The SOTA method for comparison is proposed in 2023. Are there any methods proposed in 2024 or 2025 in the ASSR field? Please provide a comparison of the results.

---

### Official Review · Reviewer_mTgg · 2025-10-30

**Soundness:** 3
**Presentation:** 4
**Contribution:** 4
**Rating:** 6
**Confidence:** 4

**Summary:**

This paper focuses on the task of arbitrary-scale super-resolution (ASSR) and, for the first time, introduces a state space model into ASSR. To achieve explicit scale adaptation and global modeling with linear complexity, the authors propose a Scalable State Space Model (SSSM) with adjustable sampling steps and transition matrices, along with a scale-aware self-attention mechanism. Experimental results validate the effectiveness of the proposed method.

**Strengths:**

1. The paper is the first to formalize an SSM as a scale-modulated SSSM for arbitrary-scale super-resolution.
2. The method is comprehensively evaluated on both synthetic (DIV2K) and real-world (COZ) benchmarks.
3. The paper presents the continuous-to-discrete formulation and the scale modulation mechanism, and provides key implementation details in the pseudocode.
4. The ablation studies offer a comprehensive analysis demonstrating the effectiveness of the SSSM and the dual-branch design.

**Weaknesses:**

1. The paper lacks discussion of extreme conditions, such as failure cases under blur, noise, or other degradation types.
2. The analysis of model efficiency does not include actual inference time measurements.

**Questions:**

1. Although the paper reports PSNR, SSIM, and LPIPS scores to demonstrate quantitative performance, these metrics mainly reflect fidelity or feature-level similarity with reference HR images. It is recommended to supplement the evaluation with no-reference image quality metrics (e.g., NIQE, BRISQUE, or PIQE) to provide a more comprehensive assessment.

2. Please provide or discuss failure cases under different or mixed degradations, such as noise, compression, or blur, to better understand the robustness of the proposed method.

---

### Official Review · Reviewer_Dm92 · 2025-10-30

**Soundness:** 3
**Presentation:** 3
**Contribution:** 2
**Rating:** 4
**Confidence:** 4

**Summary:**

The paper proposes S$^3$Mamba for arbitrary-scale SR that utilizes SSSM to replace the MLP/Conv/Attn for continuous representation learning and pixel prediction for HR image. The key innovation is to use Mamba as a substitute for MLP,  and it hasn't made a theoretical improvement to the local implicit function. The experiments demonstrate a favorable trade-off between SSM and existing methods.

**Strengths:**

- Using SSM for continuous representation learning is sound and intuitive to achieve better performance in SR task.
- The motivation is clear.
- The paper is easy to follow.

**Weaknesses:**

- The theoretical analysis and improvement are limited. It is more like an engineering validation of utilizing SSM in ASSR, as many recent Mamba-based works. I stand in a negative position for this module replacement work, like using SSM to replace attention, using attention to replace convolution/MLP, providing little new insight for this sub-area.
- Similar to existing Mamba-based work, only parameters and MACs are compared, while ignoring comparisons for the practical metrics (inference time/memory), making its efficiency unproven.
- The qualitative and quantitative improvements over the existing method are limited.
- The experiment is insufficient, for example, the real degradation comparison for LMI, and the testset contains only COZ and DIV2K.

**Questions:**

See Weaknesses.

---

### Official Review · Reviewer_JSkP · 2025-10-31

**Soundness:** 2
**Presentation:** 2
**Contribution:** 2
**Rating:** 4
**Confidence:** 5

**Summary:**

This paper proposes a novel arbitrary-scale super-resolution method, called $\text{S}^3$Mamba, to construct a scalable continuous representation space, achieving state-of-the-art (SOTA) performance.

**Strengths:**

The paper introduces State Space Models (SSM) into arbitrary-scale super-resolution, enabling scalable and continuous representation modeling with linear computational complexity.

**Weaknesses:**

1）The novelty of this paper needs to be further improved.

2）The readability of the paper is poor, and the logical coherence of writing requires further refinement.

3）More STOA baselines need to be included.

4）Analysis of theoretical depth is insufficient.

**Questions:**

1）Many existing studies have already introduced SSM into arbitrary-scale super-resolution tasks, such as MambaSR. The authors should be more rigorous when writing their contributions.

2）There are numerous spelling errors throughout the paper, including some particularly serious ones in the title and section headings. For instance, the title misspells “SCALEABLE” (which should be “SCALABLE”), a section heading reads “INTRODUTION” (should be “INTRODUCTION”), and the main text includes “discredited” (should be “discretized”). In addition, there are many punctuation and spacing errors that need to be carefully corrected.

3）The theoretical analysis is not deep enough. Most of the descriptions and formulas simply describe the execution steps of the network. For example, there is no in-depth theoretical explanation of why SSSM can learn continuous image representations, preventing readers from gaining a deeper understanding of the mechanism behind the proposed module.

4）The logical consistency of writing is weak and contains flaws. For instance, in the sentence “This attention map, guided by the current scale and coordinates, adaptively refines HR feature $F_{HR}$”, this is the first time $F_{HR}$ is mentioned, yet the authors do not explain how $F_{HR}$ is obtained, which weakens the logical clarity and readability of the paper.

5）The paper did not provide a detailed explanation of the role of SSSM, and it remains unclear what kind of mapping the SSSM is intended to learn. Is the output of the SSSM the same size as the LR image or the SR image? If it corresponds to the LR image size, then the SSSM cannot learn the LR–SR mapping relationship, nor can it model the continuous representation of the image. In that case, the SSSM would merely function as an attention mechanism, which significantly weakens the paper’s novelty.

6）More advanced and classic baseline methods should be included, such as HIIF, GaussianSR, and MambaSR, even newly proposed methods from 2025, including GSASR, Pixel to Gaussian, and Arbitrary-Scale 3D Gaussian Super-Resolution. Furthermore, the authors should add comparative experiments on benchmark datasets such as Set5, Set14, B100, Urban100, and Manga109.

7）The research motivation of this paper includes “the high computational complexity and insufficient long-range dependency modeling of existing methods.” However, the paper does not include comparisons or analyses of computational complexity (FLOPs) or inference time (Runtimes). Since this is a key part of the claimed contribution, it is strongly recommended that the authors provide a supplementary quantitative analysis.

---

### Note · Authors · 2025-11-14

I have read and agree with the venue's withdrawal policy on behalf of myself and my co-authors.